# Metabolic Syndrome but Not Fatty Liver-Associated Genetic Variants Correlates with Glomerular Renal Function Decline in Patients with Non-Alcoholic Fatty Liver Disease

**DOI:** 10.3390/biomedicines10030720

**Published:** 2022-03-19

**Authors:** Francesco Baratta, Laura D’Erasmo, Alessia Di Costanzo, Ilaria Umbro, Daniele Pastori, Francesco Angelico, Maria Del Ben

**Affiliations:** 1Department of Clinical, Internal, Anesthesiologic and Cardiovascular Sciences, Sapienza University of Rome, 00161 Rome, Italy; francesco.baratta@uniroma1.it (F.B.); daniele.pastori@uniroma1.it (D.P.); maria.delben@uniroma1.it (M.D.B.); 2Department of Translational and Precision Medicine, Sapienza University of Rome, 00161 Rome, Italy; laura.derasmo@uniroma1.it; 3Department of Public Health and Infectious Diseases, Sapienza University of Rome, 00161 Rome, Italy; francesco.angelico@uniroma1.it

**Keywords:** non-alcoholic fatty liver disease, chronic kidney disease, renal function, glomerular filtration rate, kidney, *PNPLA3*, *TM6SF2*, *MBOAT7*, *GCKR*

## Abstract

The association between non-alcoholic fatty liver disease (NAFLD) and chronic kidney disease (CKD) has been extensively demonstrated. Recent studies have focused attention on the role of patatin-like phospholipase domain-containing 3 (*PNPLA3*) rs738409 polymorphism in the association between NAFLD and CKD in non-metabolic adults and children, but the genetic impact on NAFLD-CKD association is still a matter of debate. The aim of the study was to investigate the impact of *PNPLA3*, transmembrane 6 superfamily member 2 (*TM6SF2*), membrane-bound O-acyltransferase domain containing 7 (*MBOAT7*) and glucokinase regulatory protein (*GCKR*) gene variants rather than metabolic syndrome features on renal function in a large population of NAFLD patients. The present study is a post hoc analysis of the Plinio Study (ClinicalTrials.gov: NCT04036357). *PNPLA3*, *TM6SF2*, *MBOAT7* and *GCKR* genes were analyzed by using real-time PCR with TaqMan probes. Glomerular filtration rate (GFR) was estimated with CKD-EPI. We analyzed 538 NAFLD; 47.2% had GFR < 90 mL/min/1.73 m^2^ while 5.9% had GFR < 60 mL/min/1.73 m^2^. The distribution of genotypes was superimposable according to GFR cut-offs. Results from the multivariable regression model did not show any correlation between genotypes and renal function. Conversely, metabolic syndrome was highly associated with GFR < 90 mL/min/1.73 m^2^ (odds ratio (OR): 1.58 [1.10–2.28]) and arterial hypertension with GFR < 60 mL/min/1.73 m^2^ (OR: 1.50 [1.05–2.14]). In conclusion, the association between NAFLD and CKD might be related to the shared metabolic risk factors rather than the genetic NAFLD background.

## 1. Introduction

Non-alcoholic fatty liver disease (NAFLD) is the most common liver disorder worldwide and is becoming a growing global health problem [1]. NAFLD is the major cause of end-stage liver disease but is also associated with numerous metabolic comorbidities such as obesity, type 2 diabetes, hyperlipidemia, hypertension, sleep apnea and metabolic syndrome (MetS), increasing the risk for cardiovascular disease (CVD) [2,3,4,5,6].

Recently, it has been demonstrated that NAFLD patients are also at increased risk for chronic kidney disease (CKD) [7]. The risk apparently rises with the worsening of NAFLD severity [8]. The pathogenesis of the association between NAFLD and CKD could be multifactorial and has not been completely clarified. NAFLD is often considered the main hepatic manifestation of metabolic syndrome (MetS) [9]. In addition, MetS has been clearly associated with CKD [10] and might represent the link between NAFLD and CKD [11]. However, some data have suggested that the association between NAFLD and CKD might be independent of traditional cardiovascular risk factors [12,13].

Beyond metabolic risk factors, NAFLD pathogenesis shows a heritable underpinning, with some genes involved in the pathophysiology of liver fat accumulation and its hepatic consequences [14], but not in the associated atherosclerotic damage [15]. Recent data suggested that one of these genes, the nucleotide polymorphism in the patatin-like phospholipase domain-containing protein-3 (*PNPLA3*), might be involved in the association between CKD and NAFLD. The *PNPLA3* rs738409 C > G single nucleotide polymorphism is the most common gene variant associated with NAFLD [16]. Recent evidence suggested that the *PNPLA3* I148M variant might be associated with a reduction in the estimated glomerular filtration rate (eGFR) or CKD development irrespective of common renal risk factors and presence or severity of NAFLD in both adults and children [17,18,19]. However, other studies have advocated for this association, suggesting that the genetic background might not influence the eGFR reduction, which in turn might be linked to the overlapping of cardiometabolic factors [20].

To complicate this scenario, beyond *PNPLA3*, other single nucleotide polymorphisms (SNPs) have been shown to be associated with the development of NAFLD. Among these, the stronger association was found with the rs58542926 in transmembrane 6 superfamily member 2 (*TM6SF2*) gene and, less consistently, with the rs1260326 in the glucokinase regulatory protein (*GCKR*) and the rs641738 in the membrane-bound O-acyltransferase domain containing 7 (*MBOAT7*) genes [16]. To date, poor data are available on the effect of these genes on eGFR [20].

As the impact of hepatic fat accumulation per se on CKD is still a matter of debate, we aim to investigate the influence of *PNPLA3*, *TM6SF2*, *MBOAT7* and *GCKR* genetic variants on renal function in a large population of adult NAFLD patients.

As the impact of hepatic fat accumulation per se on CKD is still a matter of debate, we aim to investigate whether NAFLD-associated gene variants in *PNPLA3*, *TM6SF2*, *MBOAT7* and *GCKR* genes or the features of metabolic syndrome affect the decline of renal function in a large population of adult NAFLD patients.

## 2. Materials and Methods

The present analysis is a post hoc analysis of the Plinio Study. (N. ClinicalTrials.gov Identifier: NCT04036357, Last Access 18 March 2022). In brief, the Plinio study is an observational prospective study aimed at investigating biochemical and pharmacological factors associated with fibrosis progression, identified as variations in noninvasive fibrosis scores, and the study of the association between NAFLD and CVD in a large population of patients with an ultrasonography diagnosis of fatty liver disease. Overall, 538 consecutive NAFLD patients with available renal function data, screened at the Day Service of Internal Medicine of the Policlinico Umberto I University Hospital in Rome, were included in the present analysis. All patients provided signed written informed consent at entry. The study protocol was approved by the local ethical board of Sapienza University of Rome (Ref. 2277 prot. 873/11) and the study was carried out according to the principles of the Declaration of Helsinki.

### 2.1. Clinical and Laboratory Workup

The inclusion criteria of the Plinio study are the presence of at least one of the following cardiometabolic diseases: arterial hypertension, overweight/obesity, type 2 diabetes, dyslipidemia, atrial fibrillation (AF) or metabolic syndrome (MetS). The exclusion criteria are: history of current or past excessive alcohol drinking as defined by an average daily consumption of alcohol > 20 g; presence of hepatitis B surface antigen and antibody to hepatitis C virus; history and clinical, biochemical and ultrasound findings consistent with cirrhosis and other chronic liver diseases; history of prior CVD and no current supplementation with vitamin E and other antioxidants. Among the Plinio population, only patients without known present or past kidney disease and who were diagnosed with NAFLD were included in the present sub-analysis.

All patients underwent a complete physical examination and ultrasonographic evaluation for liver steatosis (see below). Type 2 diabetes, hypertension as well as pharmacological treatments have been retrieved. The presence of arterial hypertension [21] and type 2 diabetes [22] was defined following international guidelines. Patients were defined as affected by MetS if more than three criteria were present [23]. Glomerular filtration rate was estimated according to the CKD Epidemiology Collaboration (CKD-EPI) equation [24] and diagnosis of CKD was performed on the basis of the more recent Kidney Disease: Improving Global Outcomes (KDIGO) guideline [25].

All ultrasound scanning (US) was performed by the same operator, who was blinded to laboratory values using a GE VividS6 apparatus equipped with a convex 3.5 MHz probe to reduce the discrepancies related to ultrasound techniques. The US was used to assess the degree of liver steatosis. Liver steatosis was defined according to Hamaguchi criteria based on the presence of abnormally intense, high-level echoes arising from the hepatic parenchyma; liver–kidney difference in echo amplitude; echo penetration into deep portions of the liver; and clarity of liver blood vessel structure [26].

Laboratory tests were performed on early-morning blood samples collected after an overnight fast in EDTA-containing tubes. Samples were delivered to the laboratory within 2 h. Tubes were centrifuged at 3000 rpm at 4 °C for 10 min to separate the plasma and buffy coat layers. The latter were stored at −80 °C and used later for genotype analysis.

Liver enzymes (alanine transaminase (ALT), aspartate transferase (AST), gamma glutamyl transferase (γGT)), HCV and hepatitis B virus antibodies, hematology, serum creatinine and coagulation were determined using standard procedures. Serum cholesterol (total and HDL fraction) and triglycerides (TG) were measured with an Olympus AN 560 apparatus using an enzymatic colorimetric method. Low-density lipoprotein (LDL) cholesterol levels were calculated according to the Friedewald formula. A Roche/Hitachi COBAS CE 6000 analyzer was used to determine plasma glucose and plasma insulin levels. The former was measured with the hexokinase/glucose-6-phosphate dehydrogenase (HK/G6P-DH) method (Roche Diagnostic GmbH, Mannheim, Germany) adapted for use with the COBAS CE 6000; the latter was measured by electrochemiluminescence immunoassay (Elecsys Insulin–Roche Diagnostic GmbH, D-68298 Mannheim) [14]. Using laboratory parameter, FIB4 index was used for estimating the severity of fibrosis in NAFLD patients [27]; a positive FIB−4 was defined as FIB−4 > 2.67 while a negative FIB−4 was determined as FIB−4 < 1.30 in patients younger than 65 years or as FIB−4 < 2.0 in patients over 65 years [28]. DNA was extracted from peripheral blood, as reported elsewhere. The rs641738 C > G (I148M) (*PNPLA3*), rs58542926 C > T (E167K) (*TM6SF2*), rs1260326 C > T (L446P) (*GCKR*) and rs641738 C > T (G17E) (*MBOAT7-TMC4*) were genotyped by TaqMan 5′-Nucleotidase assay by using ABI PRISM 7900 HT Sequence Detection System (Applied Biosystems) [16,29]. The TaqMan assays were validated by direct resequencing of representative samples of DNA on an ABI PRISM 3130 XL Genetic Analyzer. Allele frequencies were in Hardy–Weinberg equilibrium in all subject groups. The genetic risk score (GRS) was calculated based on the four selected SNPs, as previously described [16]. Both described methods were used to calculate the GRS: a simple count method (unweighted GRS) and a weighted method (weighted GRS) [17].

### 2.2. Statistical Analysis

Categorical variables were reported as counts (percentages) and continuous variables as means ± standard deviation (SD) or median and interquartile range (IQR) unless otherwise indicated. Categorical variables were tested by the chi-square test. Normal distribution of parameters was assessed by Kolmogorov–Smirnov test. The Student’s unpaired *t*-test was used for normally distributed continuous variables, the Mann–Whitney U test for the non-normally distributed ones. 

The association between NAFLD and CKD was estimated by considering two endpoints: (1) mild eGFR reduction defined as an eGFR below 90 mL/min/1.73 m^2^; and (2) moderate eGFR reduction defined as an eGFR below 60 mL/min/1.73 m^2^. Therefore, the analysis was conducted comparing: (1) NAFLD patients with mild eGFR reduction to those with normal eGFR; and (2) NAFLD patients with moderate eGFR reduction to those without it.

Then, genotype frequencies were assessed for Hardy–Weinberg equilibrium using the goodness-of-fit χ2 test. The four SNPs were tested using additive, dominant and recessive genetic models. We also conducted a sensitivity analysis by calculating alternative GRS by excluding one genetic variant at a time to check for the robustness of associations between the four NAFLD-associated risk alleles and the decline of eGFR. The dominant model of inheritance for the *PNPLA3* gene variant was chosen as the best in our cohort to estimate associations with eGFR values. In this analysis, the presence of *PNPLA3* rs738409 CG + GG and CC genotypes was combined with covariates that emerged as potential confounding factors because of their significance in univariable regression analyses or their biological plausibility. Multivariable logistic regression analyses were performed to assess factors associated with mild eGFR reduction and moderate eGFR reduction. All dichotomous variables were included in the models. Model A included MetS diagnosis and PNPLA3 genotype. Model B included PNPLA3 genotype and all the components of MetS score: hypertension, hypertriglyceridemia, hyperglycemia, low HDL cholesterol and increased waist circumference. In Model C, arterial hypertension and diabetes diagnosis substitute for the “high blood pressure” and “high blood glucose” items of the MetS score, respectively. Model D was similar to Model A but included the weighted GRS instead of PNPLA3 genotype. Collinear variables and variables used for the eGFR calculation were excluded from the multivariable analysis. Appendix A analyses were performed, adding age as a covariate.

The area under the receiver operating characteristic curve (AUROC) and its 95% C.I. were calculated for unweighted and weighted GRS on both outcomes.

Propensity score matching (PSM) was performed to match patients with *PNPLA3* GG genotype to patients with *PNPLA3* CC genotype according to age, sex, MetS and arterial hypertension. Match tolerance was set to 0.005, without replacement.

All statistical analyses were performed using SPSS software version 27.0 (SPSS Inc., Chicago, IL, USA). A 2-sided *p* value < 0.05 was considered to be statistically significant.

## 3. Results

We included 538 patients in the analysis; 38.5% (207/538) were female. The mean age was 54.5 ± 11.6 years and the mean BMI was 30.4 ± 5.0 kg/m^2^. The median eGFR in the whole cohort was 91.5 [79.4–101.1] mL/min/173 m^2^ and the prevalence of patients with mild eGFR reduction (eGFR < 90 mL/min/1.73 m^2^) was 47.2%, while only 5.9% of patients had moderate eGFR reduction (eGFR < 60 mL/min/1.73 m^2^). We found no difference in median eGFR values according to PNPLA3, TM6SF2, MBOAT7 and GCKR genotypes (Figure 1). Relatedly, the distribution of all genotypes was superimposable in the different eGFR subpopulations (Appendix A).

The prevalence of hypertension was higher considering both groups of NAFLD patients with mild and moderate eGFR reduction (64.2% vs. 54.9%, *p* = 0.029 in mild; 78.1% vs. 58.1%, *p* = 0.025 in moderate). In addition, patients with mild eGFR reduction also showed higher prevalence of high blood pressure according to MetS definition as compared with those with normal eGFR (75.2% vs. 67.3%, *p* = 0.048). Patients with mild eGFR reduction also exhibited a higher frequency of MetS as compared to others (63.0% vs. 54.6%, *p* = 0.048) (Table 1).

At univariable , mild eGFR reduction correlated with MetS (odds ratio (OR)) and 95% of confidence interval (C.I.) for OR: 1.42 (1.00–2.00)), high blood pressure (OR: 1.48 (1.01–2.15)) and arterial hypertension (Table 2, panel A). Instead, moderate eGFR reduction correlated only with arterial hypertension (OR: 2.57 (1.09–6.06)). (Table 2, panel B). After adjustment for confounders, MetS (aOR: 1.58 (1.10–2.28)) and arterial hypertension (aOR: 1.50 (1.05–2.14)) correlated with mild eGFR reduction (Table 2, panel A) while only arterial hypertension (2.79 (1.16–6.68)) correlated with moderate eGFR reduction (Table 2, panel B). On the contrary, none of the four gene variants and genotypes or the calculated weighted/unweighted GRS, nor the univariable or the multivariable analyses correlated with mild or moderate eGFR reduction (Table 2 and Appendix A).

It must be noted that to explore the presence of a specific weighted or unweighted GRS cut off that could accurately identify patients with mild or moderate eGFR reduction, we also performed ROC analyses. However, AUROCs were not significant (Appendix A). We further conducted an additional multivariable analysis, including age as covariate, confirming previous results (Appendix A). No significant correlation between PNPLA3 genotype, GRS score and eGFR was found. It is noteworthy that, in the same analysis, after including age as covariate, the associations between mild GFR reduction and high blood pressure as well as between moderate GFR and hypertension were lost (Appendix A). This effect might be driven by the high correlation between age and MetS (OR: 1.04 (1.02–1.05)) and arterial hypertension (OR: 1.07 (1.05–1.09)) (Appendix A).

To further confirm our results, we tested the propensity score matching methods including sex, age, MetS and arterial hypertension as covariates. Sixty-four patients with PNPLA3 GG genotype were matched with 64 patients with PNPLA3 CC genotype without finding any difference in median eGFR values (GG vs. CC: 93.7 (82.0–104.8) vs. 94.6 (81.2–104.2) mL/min/1.73 m^2^, *p* = ns) (Appendix A).

## 4. Discussion

Nonalcoholic fatty liver disease is associated with an increased risk of CKD in adults. However, it is uncertain whether this association is influenced by NAFLD susceptibility gene variants or metabolic disorder load characterizing NAFLD patients. To our knowledge, this is the first study investigating the relationship between the four major NAFLD risk alleles in *PNPLA3*, *TM6SF2*, *GCKR* and *MBOAT7* genes and kidney function in a large population of 538 adult patients. We demonstrated that NAFLD susceptibility variants had no impact on eGFR decline and MetS was the major factor independently associated with impaired GFR.

Previous studies described a possible link between *PNPLA3* gene variant and renal function in patients with or without NAFLD. Most of these studies were performed in children with NAFLD. Targher and colleagues [17] found that in 142 children with biopsy-proven NAFLD, those with *PNPLA3* GG genotype had lower eGFR (107.5 ± 20 vs. 112.8 ± 18 vs. 125.3 ± 23 mL/min/1.73 m^2^, *p* = 0.002) compared to those with CG and CC genotypes, respectively. However, these patients showed a very high prevalence of *PNPLA3* G allele which probably represented the main pathophysiological cause of their severe liver disease. Later, these data were confirmed by Marzuillo et al. [30], who found a reduction in eGFR in obese children. However, the association persisted after adjustment for metabolic risk factor only in patients with NAFLD. These findings are in line with the recent work by Di Costanzo and colleagues [20] which found an association between *PNPLA3* G risk allele and eGFR in obese children with NAFLD, even after adjustment for BMI and hepatic fat fraction. However, when authors tested the association between the NAFLD susceptibility gene variant and eGFR reduction in the whole cohort, NAFLD was the only factor associated with GFR reduction, thus demonstrating that fatty liver and the shared metabolic risk factors have a stronger effect on the decline of renal function than the PNPLA3 G risk allele. Interestingly, in this study, *TM6SF2, GCKR* and *MBOAT7* variants did not show any impact on kidney function. While data are substantially homogeneous in children, studies are contradictory in adults. In two different studies performed in elderly patients (mean age around 70 years) with type 2 diabetes, Mantovani and colleagues [19,31] found lower eGFR and higher prevalence of CKD in those carrying the *PNPLA3* rs738409 polymorphism, independently from NAFLD. By contrast, in studies conducted in younger adult populations, the association was weaker or missing. Dan-Qin Sun et al. [32] found no association between *PNPLA3* genotypes in their whole population of 217 histologically proven NAFLD but only in the subgroup of 75 patients with persistently normal ALT. Finally, Yuya Seko et al. [33], in the largest cohort previously investigated of biopsy-proven NAFLD patients (*n* = 344), found that *PNPLA3* genotypes were not associated with baseline eGFR nor with eGFR decline and the development of CKD at the end of follow-up. Our data are in accordance with those from Yuya Seko and Dan-Qin Sun.

As in their study, patients included in the present analysis are almost middle aged, different from the older ones included in the study from Mantovani [19,31]. In addition, Mantovani and colleagues evaluated cohorts composed exclusively of diabetic patients and results could not necessarily be extended to all NAFLD population, diabetes being only one of the risk factors for liver steatosis. In addition, one of these studies was performed only in post-menopausal women, amplifying the possible confounders.

Unlike previous studies, we also investigated, in an adult population, the combined effect of all NAFLD genetic risk variants on renal function. The weighted GRS combining *PNPLA3*, *GCKR* and *TM6SF2* risk alleles has already been associated with almost eightfold higher risk of NAFLD in obese children, being a useful tool to predict liver fat accumulation. According to what was found by Di Costanzo et al. [20], the NAFLD GRS did not show any detrimental effect on renal function in obese children.

In our study, we found that MetS and arterial hypertension, among its components, are the only factors independently associated with impaired GFR. These findings confirm, firstly in NAFLD patients, what was previously observed in the general population [34,35,36].

We speculate that the association between *PNPLA3* rs738409 polymorphism and eGFR found in children might be a consequence of the higher allele variant frequency and the lower exposure time to metabolic risk factors in NAFLD children. Conversely, environmental risk factors for NAFLD become prevalent in adults, as demonstrated by the association we found between MetS and patients’ age, influencing renal function in turn. Based on these considerations, we cannot exclude that the development of CKD in NAFLD patients can be multifactorial, being determined by the accumulation in the same patient of both metabolic and genetic risk factors, especially with the older age that we know to be one of the strongest determinants of eGFR decline.

Our study has some strengths. We carefully defined patients according to the four major NAFLD susceptibility gene variants, both considering each polymorphism alone and including them in a genetic risk score. In addition, compared to other studies, our study population is more representative of the complex physiopathology of NAFLD in adult patients with diversified metabolic profile and demographic characteristics.

Our study also has some limitations that should be mentioned. First, NAFLD diagnosis was performed by ultrasonography which does not represent the gold standard for NAFLD diagnosis. However, due to the high invasiveness of liver biopsy, US is the more widely used technique for the screening of NAFLD in routine practice. In addition, the US has a specificity of around 100% for NAFLD [1] and our study includes only patients with US-NAFLD.

Second, we did not measure GFR but it is estimated by a validated equation. In addition, we did not retrieve data on albuminuria so that CKD diagnosis, due to the post hoc design of the study, can be estimated only when an eGFR is below 60 mL/min/1.73 m^2^, but we cannot exclude the presence of albuminuria > 30 mg/g in some patients with eGFR above 60 mL/min/1.73 m^2^. Finally, we cannot exclude residual confounding as a result of unmeasured risk factors.

## 5. Conclusions

Our results show that in a large cohort of NAFLD adult patients, the major contribution to eGFR decline is the presence of metabolic syndrome, mainly arterial hypertension. None of the NAFLD-associated genetic risk variants contributed to eGFR decline, suggesting that in adult patients, metabolic risk factors are more important that genotype for renal function changes.

## Figures and Tables

**Figure 1 biomedicines-10-00720-f001:**
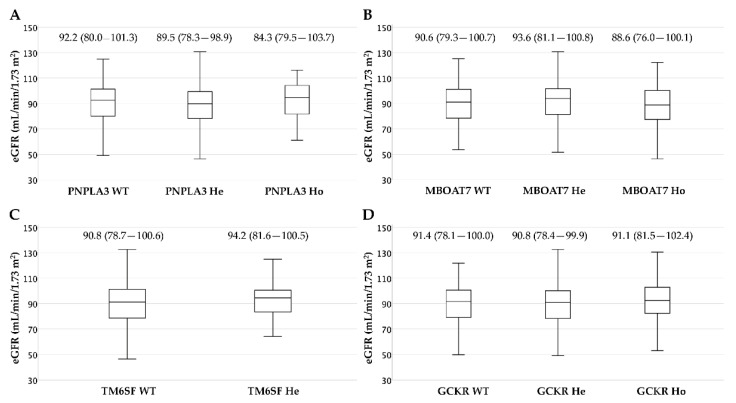
Box plots of eGFR according to PNPLA3 (Panel **A**), MBOAT7 (Panel **B**), TM6SF2 (Panel **C**) and GCKR (Panel **D**) genotypes.

**Table 1 biomedicines-10-00720-t001:** Patients’ characteristics according to reduced eGFR.

Variables	eGFR ≥ 90 mL/min/1.73 m^2^ (*n* = 284)	eGFR < 90 mL/min/1.73 m^2^ (*n* = 254)	*p*	eGFR ≥ 60 mL/min/1.73 m^2^ (*n* = 506)	eGFR < 60 mL/min/1.73 m^2^ (*n* = 32)	*p*
Age (y)	50.7 ± 10.9	58.7 ± 10.9	<0.001	53.9 ± 11.4	64.5 ± 11.1	<0.001
Women (%)	37.7	39.4	0.687	38.5	37.5	0.907
BMI (kg/m^2^)	30.6 ± 5.3	30.1 ± 4.7	0.262	30.4 ± 5.1	30.0 ± 3.5	0.681
Blood glucose (mg/dL)	105.0 ± 30.3	105.1 ± 26.8	0.964	105.2 ± 28.8	102.3 ± 26.3	0.578
High blood glucose (%) *	43.7	52.0	0.054	48.0	40.6	0.416
Type II Diabetes (%)	27.5	29.5	0.596	28.7	25.0	0.657
Waist circumference (cm)	107.7 ± 12.5	106.6 ± 11.1	0.285	107.3 ± 12.1	106.6 ± 8	0.684
High waist circumference (%) *	79.9	81.1	0.732	80.2	84.4	0.567
High blood pressure (%) *	67.3	75.2	0.043	70.8	75.0	0.607
Arterial hypertension (%)	54.9	64.2	0.029	58.1	78.1	0.025
HOMA-IR	3.2 (2.3–5.0)	3.4 (2.5–5.6)	0.086	3.3 (2.3–5.5)	3.2 (2.4–4.5)	0.471
Total cholesterol (mg/dL)	201.4 ± 40.8	198.4 ± 39.9	0.393	200.5 ± 40.9	192.2 ± 30.5	0.261
HDL cholesterol (mg/dL)	48.0 ± 13.2	47.7 ± 14.7	0.805	47.9 ± 14.1	48.0 ± 12.4	0.971
Low HDL cholesterol (%) *	39.8	38.2	0.704	39.1	37.5	0.855
LDL cholesterol (mg/dL)	121.6 ± 36.1	120.1 ± 34.3	0.603	121.3 ± 35.6	114.9 ± 28.6	0.323
Triglycerides (mg/dL)	136.0 (100.0–186.0)	135.0 (110.0–170.0)	0.766	57.9	68.8	0.227
High triglycerides (%) *	41.5	42.9	0.749	41.5	53.1	0.197
Metabolic syndrome (%) *	54.6	63.0	0.048	57.9	68.8	0.227
GGT (IU/L)	28.0 (18.0–49.5)	27.0 (18.0–40.0)	0.123	28.0 (18.0–43.0)	28.0 (17.0–43.7)	0.866
ALT (IU/L)	30.0 (20.0–46.5)	26.0 (19.0–39.0)	0.029	28.0 (20.0–43.0)	25.0 (15.5–33.2)	0.075
AST (IU/L)	22.0 (18.0–29.0)	21.0 (18.0–27.0)	0.517	21.0 (18.0–28.0)	21.0 (16.5–30.7)	0.962
FIB4− (%)	84.2	82.3	0.561	83.8	75.0	0.196
FIB4+ (%)	2.1	2.8	0.628	2.6	0	0.359
Severe steatosis (%)	32.4	37.3	0.234	35.5	21.9	0.116

* According to ATP III modified criteria [24].

**Table 2 biomedicines-10-00720-t002:** Univariable and multivariable analyses of factors associated with reduced GFR.

**Panel A. Factors Associated with eGFR < 90 mL/min/1.73 m^2^**					
**Variables**	**Univariable Analysis** **OR** **(95% C.I.)**	**Model A** **OR** **(95% C.I.)**	**Model B** **OR** **(95% C.I.)**	**Model C** **OR** **(95% C.I.)**	**Model D** **OR** **(95% C.I.)**
BMI	0.98 (0.95–1.01)	0.97 (0.93–1.01)	-	-	0.97 (0.93–1.01)
Metabolic syndrome	1.42 * (1.00–2.00)	1.58 * (1.10–2.28)	-	-	1.58 * (1.10–2.27)
FIB4−	0.87 (0.56–1.38)	0.89 (0.56–1.41)	0.94 (0.59–1.50)	0.92 (0.58–1.47)	0.88 (0.56–1.40)
PNPLA3 CG/GG	1.16 (0.83–1.63)	1.12 (0.79–1.59)	1.19 (0.84–1.68)	1.20 (0.85–1.71)	-
High blood glucose ^#^	1.40 (0.99–1.96)	-	1.32 (0.92–1.68)	-	-
Diabetes	1.11 (0.76–1.61)	-	-	0.99 (0.67–1.47)	-
High waist circumference ^#^	1.08 (0.70–1.65)	-	0.97 (0.62–1.52)	1.04 (0.67–1.61)	-
High blood pressure ^#^	1.48 * (1.01–2.15)	-	1.41 (0.96–2.08)	-	-
Arterial hypertension	1.47 * (1.04–2.08)	-	-	1.50 * (1.05–2.14)	-
Low HDL cholesterol ^#^	0.93 (0.66–1.32)	-	0.91 (0.62–1.32)	0.87 (0.60–1.27)	-
High triglycerides ^#^	1.06 (0.75–1.49)	-	1.07 (0.73–1.55)	1.12 (0.77–1.63)	-
Weighted GSR	1.24 (0.64–2.39)	-	-	-	1.19 (0.61–2.33)
**Panel B. Factors Associated with eGFR < 60 mL/min/1.73 m^2^**					
**Variables**	**Univariable Analysis** **OR** **(95% C.I.)**	**Model A** **OR** **(95% C.I.)**	**Model B** **OR** **(95% C.I.)**	**Model C** **OR** **(95% C.I.)**	**Model D** **OR** **(95% C.I.)**
BMI	0.98 (0.91–1.06)	0.96 (0.89–1.05)	-	-	0.96 (0.89–1.04)
Metabolic syndrome	1.60 (0.74–3.45)	1.72 (0.77–3.85)	-	-	1.72 (0.77–3.84)
FIB4−	0.58 (0.25–1.34)	0.57 (0.24–1.33)	0.54 (0.23–1.28)	0.60 (0.25–1.41)	0.57 (0.24–1.33)
PNPLA3 CG/GG	0.98 (0.48–2.01)	0.91 (0.44–1.90)	1.01 (0.48–2.11)	1.09 (0.52–2.28)	-
High blood glucose ^#^	0.74 (0.36–1.53)	-	0.59 (0.27–1.28)	-	-
Diabetes	0.83 (0.36–1.89)	-	-	0.62 (0.260–1.45)	-
High waist circumference ^#^	1.33 (0.50–3.54)	-	1.40 (0.51–3.87)	1.26 (0.46–3.46)	-
High blood pressure ^#^	1.24 (0.54–2.82)	-	1.31 (0.56–3.04)	-	-
Arterial hypertension	2.57 * (1.09–6.06)	-	-	2.79 * (1.16–6.68)	-
Low HDL cholesterol ^#^	0.93 (0.45–1.95)	-	0.80 (0.36–1.77)	0.70 (0.32–1.57)	-
High triglycerides ^#^	1.60 (0.78–3.27)	-	1.77 (0.82–3.86)	1.82 (0.83–3.98)	-
Weighted GSR	0.92 (0.23–3.65)	-	-	-	0.80 (0.20–3.28)

^#^ According to ATP III modified criteria [24]; * *p* < 0.05. Model A: including BMI, metabolic syndrome, PNPLA3 GG/CG and FIB4−. Model B: including components of metabolic syndrome (namely, high blood glucose, high waist circumference, high blood pressure, low HDL cholesterol, high triglycerides) instead of the composite score, PNPLA3 GG/CG genotype and FIB4−. Model C: including arterial hypertension instead of high blood pressure, diabetes instead of high blood glucose, high waist circumference, low HDL cholesterol, triglycerides, PNPLA3 GG/CG genotype and FIB4−. Model D: including BMI metabolic syndrome, weighted GRS and FIB4−.

## Data Availability

The data presented in this study are available on request from the corresponding author. The data are not publicly available due to privacy.

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
