# Peer review of "Metabolic Syndrome but Not Fatty Liver-Associated Genetic Variants Correlates with Glomerular Renal Function Decline in Patients with Non-Alcoholic Fatty Liver Disease"

_biomedicines, 2022, doi:10.3390/biomedicines10030720_

Round 1

Reviewer 1 Report

Thank you for inviting me to review the revised manuscript!

Overall the paper was well written. The authors put a lot of effort to get this amount of detailed data from a large population. The analysis was reasonable and the conclusions were well supported by the results. The discussion was well organized and with in-depth explanation. Thank you for bringing the novel findings to the field!

To make the study easier understood by the readers, I have some comments:

  1. Abstract:

I notice the abstract didn’t mention the metabolic comorbidities at the beginning. I suggest the authors should talk about both the gene variation and MetS like what they have done in the introduction as MetS was concluded as a more important and relevant factor as the title highlights.

  1. Introduction:

Line 46, 47: It is not necessary to mention CVD.

I suggest removing the content not related to the topic.

In the aim, please include the investigation of the relation between MetS and renal function to match the title and conclusion.

I also suggest adding a formal hypothesis after the aims. The hypothesis is kind of an expected conclusion at the beginning of the study.

  1. Discussion:

The Discussion were well organized. The authors interpreted the results very well and reviewed and compared the results with the existing literatures. Here are some comments:

In the first paragraph,

  • I suggest to briefly summarize the background at the beginning.
  • Line 252, 253

Instead of just mentioning “In our cohort of 538 patients, 252 we found no association between NAFLD susceptibility variants and eGFR decline”, Please state the finding including the MetS to match the title and conclusions.

Author Response

March 15, 2022

Editorial Office

Biomedicines

MDPI

We would like to sincerely thank you and the Reviewer 1 for the careful revisions and comments. A point by point reply has been prepared and responses to Reviewer as well as changes in the text have been reported in red. 

Thank you for inviting me to review the revised manuscript!

Overall the paper was well written. The authors put a lot of effort to get this amount of detailed data from a large population. The analysis was reasonable and the conclusions were well supported by the results. The discussion was well organized and with in-depth explanation. Thank you for bringing the novel findings to the field!

  • We thank the reviewer for his/her thoughtful consideration of the manuscript and for the positive comments.

To make the study easier understood by the readers, I have some comments:

Abstract:

I notice the abstract didn’t mention the metabolic comorbidities at the beginning. I suggest the authors should talk about both the gene variation and MetS like what they have done in the introduction as MetS was concluded as a more important and relevant factor as the title highlights.

  • Thank you for having raised this point. The concept of Metabolic syndrome has been added in the current version of the abstract as follows: “Aim of the study was to investigate the impact of PNPLA3, Transmembrane 6 Superfamily Member 2 (TM6SF2), Membrane Bound O-Acyltransferase Domain Containing 7 (MBOAT7) and Glucokinase Regulatory Protein (GCKR) gene variants rather than metabolic syndrome features on renal function in a large population of NAFLD patients.”

Introduction:

Line 46, 47: It is not necessary to mention CVD. 

I suggest removing the content not related to the topic. 

  • According to the reviewer's suggestion we have removed the sentence pointing out CVD as the major cause of mortality/morbidity in NAFLD (please see line 48).

In the aim, please include the investigation of the relation between MetS and renal function to match the title and conclusion. 

  • According with reviewer request we have changed the aim in the introduction as follows “As the impact of hepatic fat accumulation per se on CKD is still a matter of debate, we aim to investigate whether NAFLD-associated gene variants in PNPLA3, TM6SF2, MBOAT7 and GCKR genes or the features of metabolic syndrome affect the decline of renal function in a large population of adult NAFLD patients.”

I also suggest adding a formal hypothesis after the aims. The hypothesis is kind of an expected conclusion at the beginning of the study

  • As the impact of NAFLD associated gene variants on eGFR decline is still a matter of debate, we would not expect these results as other studies have shown different results. However, as pointed out in the discussion, our approach allowed us to extend the observation on a larger population of well characterised NAFLD patients that we believe are more representative of the general population. However, to emphasise our methodological approach, we deeply described our aims, clarifying our null hypothesis (that it may be the metabolic environment more than genetic variants to affect GFR). Please see lines 80-83

Discussion:

The Discussion were well organized. The authors interpreted the results very well and reviewed and compared the results with the existing literatures.

  • We thank the reviewer for the positive comments.

Here are some comments:

In the first paragraph, 

  1. I suggest to briefly summarize the background at the beginning. 
  • To accomplish the reviewer’s request, we have briefly summarised the background at the beginning of the Discussion Section as follows: ‘Nonalcoholic fatty liver disease is associated with an increased risk of CKD in adults. However, it is uncertain whether this association is influenced by NAFLD susceptibility genes variants or metabolic disorder load characterizing NAFLD patients. To our knowledge, this is the first study investigating the relationship between the four major NAFLD risk alleles in PNPLA3, TM6SF2, GCKR, and MBOAT7 genes and kidney function in a large population of 538 adult patients. We demonstrated that NAFLD susceptibility variants had no impact on eGFR decline and MetS was the major factor independently associated with impaired GFR.”  Line 258-265

Instead of just mentioning “In our cohort of 538 patients, 252 we found no association between NAFLD susceptibility variants and eGFR decline”, Please state the finding including the MetS to match the title and conclusions. 

  • As requested we have changed the first paragraph (see above).

I look forward to hearing from you at your earliest convenience.

Yours sincerely,

Ilaria Umbro, MD, PhD

Reviewer 2 Report

  1. The table 1 is not fully visible.
  2. "In our cohort of 538 patients, we found no association between NAFLD susceptibility variants and eGFR decline." This is only a negative result, Authors were not able to find any positive correlation. 

Author Response

Editorial Office

Biomedicines

MDPI

We would like to sincerely thank you and the Reviewer 2 for the careful revisions and comments. A point by point reply has been prepared and responses to Reviewer as well as changes in the text have been reported in red. 

  1. The table 1 is not fully visible.
  • The format of the table was changed accordingly, to improve table visibility.

  1. "In our cohort of 538 patients, we found no association between NAFLD susceptibility variants and eGFR decline." This is only a negative result, Authors were not able to find any positive correlation. *
  • The lack of any association between NAFLD-associated gene variants and eGFR decline is the major result of our study. However, as we have also highlighted the presence of a clear association between eGFR decline and features of metabolic syndrome, this observation has been included in the summary of the results (discussion, first paragraph).

Round 2

Reviewer 2 Report

I do not have any additional remarks. Authors' response fulfilled all my concerns.

This manuscript is a resubmission of an earlier submission. The following is a list of the peer review reports and author responses from that submission.

Round 1

Reviewer 1 Report

Comment:

Thank you for the invitation to review this well-written paper.

Major strengths:

The authors present a valuable first study investigating the relationship between four major NAFLD risk alleles and kidney function in a large population of adult NAFLD. The detailed data and appropriate methods well supported the conclusions. The authors also compared and discussed the different findings with the previous studies. These differences resulted from various cohort populations like elderly, children, diabetic patients, and post-menopausal women.

Major weaknesses:

No major weakness.

Minor amendments:

There were some limitations of not using the golden standard for the diagnosis of NAFLD and CKD.

Overall, the authors presented a novel study with accurately interpreted the data and drew an appropriate conclusion. It is valuable for the field to understand the association between NAFLD, genetic background, and CKD.

Reviewer 2 Report

  1. To name a group with eGFR < 60 ml/min/1.73 m2 as CKD might be misleading, as far as Authors admitted that all known clinical data were only eGFR. So this group should be described as "eGFR < 60 ml/min/1.73 m2". I agree that reduced GFR is enough to define CDK but on the other hand it might happen that among those with eGFR 60 ml/min or even 90 ml/min, there are patients who will fulfil CKD criteria, in case of, for example proteinuria. 
  2. Patients assigned to lower GFR groups (both < 90 ml/min and < 60 ml/min) were significantly older (the part of CKD-EPI equation is age)
  3. The more specific clinical characteristic of included patients should be provided. 
  4. Figure 1 should be presented in more readable way.

Reviewer 3 Report

Barrata et al. attempt to identify risk factors for the development of CKD in NAFLD patients.

Unfortunately the manuscript has some limitations, which I believe limit generalizability of the study results and preclude any definitive conclusions:

  1. The group is quite homogenous considering kidney function, with a median eGFR in the whole cohort of 91.5 [79.4-101.1] mL/min/173m2; 254 patients with eGFR <90ml/min/1,73m2 and only 32 patients with eGFR <60ml/min/1,73m2. No further stratification is provided.
  2. Patients with eGFR <90 are significantly older (mean age is 8 years higher in patients with lower eGFR) than patients with eGFR >90. It is a known fact that eGFR declines with age, even in people without kidney disease. From the description of statistical methods I understand that age was excluded from the multivariable analysis, so it is impossible to tell to what extend eGFR value was influenced by physiological age-related decline of kidney function and to what extend by other analyzed factors.
  3. Also patients with eGFR <60ml/min/1,73m2 are significantly older (ca. a decade) than patients with eGFR <60ml/min/1,73m2.
  4. It is unclear how were the variables chosen to be introduced into multivariate models. Why would the Authors introduce variables which were insignificant in univariate analysis?
  5. Was there a significant correlation between metabolic syndrome and/or arterial hypertension and age?
  6. Did any of the patients had a diagnosis of a specific kidney disease? This was not listed as an exclusion criterion.